# Evaluating Energy Generation Capacity of PVDF Sensors: Effects of Sensor Geometry and Loading

**DOI:** 10.3390/ma14081895

**Published:** 2021-04-10

**Authors:** Mohammad Uddin, Shane Alford, Syed Mahfuzul Aziz

**Affiliations:** UniSA STEM, University of South Australia, Mawson Lakes, SA 5095, Australia; alfsc001@mymail.unisa.edu.au (S.A.); Mahfuz.Aziz@unisa.edu.au (S.M.A.)

**Keywords:** PVDF, piezoelectric material, energy harvesting, human body movements, power generation

## Abstract

This paper focuses on the energy generating capacity of polyvinylidene difluoride (PVDF) piezoelectric material through a number of prototype sensors with different geometric and loading characteristics. The effect of sensor configuration, surface area, dielectric thickness, aspect ratio, loading frequency and strain on electrical power output was investigated systematically. Results showed that parallel bimorph sensor was found to be the best energy harvester, with measured capacitance being reasonably acceptable. Power output increased with the increase of sensor’s surface area, loading frequency, and mechanical strain, but decreased with the increase of the sensor thickness. For all scenarios, sensors under flicking loading exhibited higher power output than that under bending. A widely used energy harvesting circuit had been utilized successfully to convert the AC signal to DC, but at the sacrifice of some losses in power output. This study provided a useful insight and experimental validation into the optimization process for an energy harvester based on human movement for future development.

## 1. Introduction

The use of portable electronics has been growing rapidly. As the emerging portable electronic devices are embedding more functionalities, they are becoming complex in nature, requiring higher amount of electrical energy for their operation. With higher levels of energy consumption, the batteries used to supply power to the portable devices deplete their energy quite quickly. Hence, there is increasing demand for alternative energy, and this has facilitated growth in the area of renewable energy sources, which can be harnessed to power portable devices ‘on the go’. Current postulated application areas for portable energy harvesting include wireless sensor networks, biomedical (e.g., pacemakers) and military use [1,2]. In particular, energy harvesting via human body movements (e.g., running, jogging) is seen as a promising means to power portable devices carried by people. In this regard, different types of piezoelectric materials such as piezoelectric (PZT), polyvinylidene difluoride (PVDF), have been studied to understand and evaluate the efficacy of electrical power generation from the mechanical energy harvested from human movements. For example, piezoelectric materials tapped on shoe sole and wearable fabrics have been found to generate power [3]. As the amount of power generated through such movements is relatively small, the selection of appropriate piezoelectric material along with design of the energy harvesting circuits is crucial to generate usable power. In the past, with the aid of experimental and simulation tools, tremendous efforts have been made to study the design, implementation and evaluation of piezo sensors with the aim of increasing the amount of power generation [4]. Furthermore, PVDF is flexible, resistant to mechanical shock and highly compatible with the environment, which make it suitable for full PZT beam without needing any substrate layer. As such, researchers focused on gross energy scavenging capacity of PVDF based sensors in different applications, e.g., human walking, elbow bending. However, little research is reported on comprehensive parametric study of PVDF sensors in a holistic way. For example, studies focusing on the effects of different sensor parameters including geometry, mechanical loading, frequency and straining on electrical power generation capability of PVDF sensors have not been reported.

This paper presents the effect of the type and geometry of PVDF (polyvinyl difluoride) piezoelectric sensors on the amount of power generated, along with power conversion circuitry to be embedded in wearable-fabrics to harvest energy via human body movements. PVDF was chosen due to its flexibility, low cost, and good piezo-properties, such as high electric charge accumulation under small mechanical strain. Three piezoelectric sensor configurations, namely-parallel bimorph, series bimorph and unimorph sensors were investigated in this study. The effect of dielectric thickness, surface area and aspect ratio, loading type and frequency on the electrical power output of each sensor configuration were studied, with the aim of optimizing the senor parameters. A simple power conversion circuit without a voltage regulator was designed and implemented, and its efficiency in terms of power output and potential issues was discussed in this paper. The results are analyzed with reference to available literature on energy harvesting.

## 2. Related Works

The human body contains a wide variety of energy sources, for example body heat, breath, blood pressure, upper limb motion, walking and finger motion [5]. It is reported that upper arm motion could produce up to 60 W of power (i.e., arm lifts above head at 1.3 lifts/s by a man weighing 58 kg), of which up to 0.33 W is recoverable using piezoelectric fabric [6], while a man weighing 68 kg walking at 2 steps/s could produce 324 W of power, with up to 8.3 W recoverable using piezoelectric inserts. These estimations are under ideal conditions and any attempt at generating electricity using these methods is likely to produce far less power due to design inefficiencies and component energy loss. With the advancement of new materials, researchers have developed new PZT thin films to increase energy harvesting capacity. For example, Won et al. studied Pb(Zr,Ti)O_3_ (PZT) thin film with a LaNiO_3_ (LNO) buffer layer on an ultra-thin Ni-Cr-based austenitic steel metal foil substrate to generate electrical power from vibrational energy [7]. They reported that the maximum power and the corresponding peak voltage generated by the sensor were 5.6 µW and 690 mV, respectively. Yeo et al. [8] developed a flexible tactile sensor array consisting of aluminum nitride (AlN) thin film as PZT material deposited in Si wafer for energy generation when the sensor is subject to mechanical stimulus.

Despite the overwhelming success with PZT, PVDF is considered to be the most commonly used piezoelectric materials [9]. PZT is too stiff (Young’s modulus of 63 GPa as opposed to 8.3 GPa for PVDF) to be of any use where flexibility is required, leaving PVDF as the best choice for use on wearable fabrics to harvest energy. Jones et al. [10] outlined the preparation of PVDF to increase the piezoelectric effect by poling and stressed that uniform crystal polarization must be achieved in order to maximise the potential power generation capabilities of the material. A PVDF coating was created by mixing, curing and poling PVDF solution and then coated onto conductive fabric to from a flexible piezo-electric membrane [2]. Paradiso and Starner [11] proposed a PVDF shell structure to increase power output, in which, the PVDF was bonded to a pre-curved polyester film and electrodes were attached on either side. The maximum voltage generated was about 40 V. The sensitivity and power output of three different types of piezo-electric actuators were investigated [12]. The three actuator configurations were: (1) parallel triple layer (parallel bimorph), (2) series triple layer (series bimorph) and (3) unimorph. A detailed illustration of the configurations is presented in [12]. Most research in the field of piezoelectric energy harvesting had focused on high frequency loading. Low frequency loading may produce different output characteristics similar to static loading. Further, Ozeri and Shmilovitz [13] analyzed the time response properties of the piezoelectric harvesters.

Using 3D simulation, a unimorph PZT generator consisting of a 200 µm thick steel substrate and 10 µm to 400 µm dielectric was investigated [14]. It was observed that the generated current increased with increasing dielectric thickness until a maximum was reached after which it decreased with increasing thickness. The voltage followed a different trend, which increased continually with increasing dielectric thickness. Granstrom et al. [15] tested the power generating capability of a PVDF “shoulder strap” to be integrated into an energy generating backpack. The average power generated was 45.6 mW, and a PVDF sample of 28 µm thickness consistently produced a greater power output than that of 56 µm sample by an average of 0.9 mW or 62%.

Waqar et al. [16] performed a dual-field computational analysis on a simulated PVDF patch bonded to a flexible fabric and studied the effect of various sensor parameters on the electrical power output. They used a PVDF film of 0.2 mm thickness, where electrodes on either side connected to a resistor “that was matched to the piezo properties” reported in [2]. It was observed that a surface area of 400 mm^2^ (with aspect ratio of 4 and 0.005 N load), aspect ratio of 8 (with surface area of 1600 mm^2^ and 0.005 N load) and load of 0.05 N (with surface area of 1600 mm^2^ and aspect ratio of 4) produced the highest power output. Only one variable was changed at a time, and no attempt was made to maximize the power output using the optimum value for all three parameters at the same time. Further, only cantilever load was utilized in this analysis, which may not reflect actual loading scenario a PVDF being subject to due to human body movements.

An adaptive energy harvesting circuit was presented to maximize the efficiency of piezoelectric energy harvesting [17]. Through testing with a Quickpack^®^ actuator element attached to a shaker operating at 53.8 Hz, the system was shown to increase power transfer significantly. The major drawback of this system is that it uses a relatively large amount of energy to power the microcontroller required for adaptive control, thus negating the benefit of renewable energy generation.

It is well known that piezoelectric devices generate electrical signals only in response to a change in the applied force, because under static stress, the free carriers drift toward the dipoles, eventually discharging the device [18]. Piezoelectric devices can be modelled by the Butterworth Van Dyke model, where a series connection of resistors (*Rs*), inductors (*Ls*) and capacitors (*Cs*) models the mechanical resonance. The design of a proper energy harvesting circuit is thus essential to minimize the loss of input energy, thus maximizing the storage of harnessed electrical energy.

The above review analysis reveals that researchers have attempted on gross energy scavenging capacity of PVDF based sensors in different applications, e.g., human walking, elbow bending. However, studies focusing on the effects of different sensor parameters, including geometry, mechanical loading, frequency and straining on electrical power generation capability of PVDF sensors have been very limited. Thus, this paper aims to propose a comprehensive experimental study in investigating the effect of sensor geometry and loading parameters on electrical power generation of PVDF sensors.

## 3. Materials and Method

### 3.1. Piezoelectric Sensor Design and Construction

Three piezoelectric sensor configurations investigated in this study were unimorph (U), parallel bimorph (PB) (parallel triple layer), and series bimorph (SB) (series triple layer). The corresponding circuit models are shown in Figure 1. A simple unimorph sensor is essentially a capacitor $C_{p}$ and resistor $R_{p}$ connected in series with a voltage source $V_{s}$ and a total output voltage of $V_{TU}$ (Figure 1a). As the name suggests, a parallel bimorph consists of two dielectrics connected in parallel, so the equivalent capacitive circuit model is two capacitors connected in parallel, effectively doubling the capacitance (Figure 1b). On the other hand, the series bimorph is two dielectrics connected in series, so the equivalent capacitive circuit model is two capacitors connected in series (Figure 1c).

The sensor prototypes were constructed using sandwiched layers of PVDF and copper shim (3M™ Red Dot™, 3M Ltd, Sydney, Australia) joined by a 2-part epoxy adhesive with the whole structure laminated (Figure 2a). PVDF used in this study is in β phase with polarization characteristics. Copper shims were used as electrodes, while the purpose of epoxy was to keep the dielectric in contact with the conductive copper. Ideally, the epoxy would be spread as thinly and evenly as possible to minimize the contact resistance and maintain uniformity across the whole sensor. The PVDF and copper were first cut to the required size (Figure 2b) and joined using epoxy. Two electrode-dielectric (copper shim-PVDF) pairs were prepared first by joining them with epoxy and placing under a uniform load of 20 kg for a period of two hours to make sure the epoxy was spread as thinly and evenly as possible. For the parallel bimorph, the two end pairs, as prepared above, were joined to a center electrode and placed under an identical load. For the series bimorph, the end pairs were joined directly together on the PVDF side. For the unimorph, a PVDF layer sandwiched between two copper shims (of 0.025 mm thick) were joined at once. Once the sensors were fabricated, the leads were soldered to the electrodes (Figure 2c). The final step was to laminate the sensor to create greater structural integrity and to provide isolation for the electrodes.

### 3.2. Sensor Geometry

For a given sensor, the effect of thickness, aspect ratio and surface area on the power output was investigated. The range of values of each parameter and the corresponding sensor type considered are summarized in Table 1, where PB, SB and U stand for parallel bimorph, series bimorph and unimorph, respectively. The aspect ratio is defined as the ratio of the length, L to the width, W of the PVDF film used in the sensor. Note that, in this study, we first aimed to demonstrate and compare the performance of three configurations for the PVDF sensor to decide which configuration gives the best power output.

### 3.3. Applied Loading and Frequency

Cantilever tensile/compressive loading was used to evaluate the power generation capabilities of the sensors. Two different forms of cantilever loading were applied to simulate the types of motions likely to be caused by the movement of the human body. One is called ‘Bending’ and the other ‘Flicking’.

‘Bending’ simulates low impact and low velocity loading. To apply this loading, the sensor was placed on a stepped surface with approximately half of the area of the sensor fastened to the top ‘step’ with a masking tape to allow very low freedom of movement (0.5 mm). The surface of the bottom step is approximately 1.5 mm below the sensor (Figure 3a). The fastening of the sensor to the top step was not completely rigid to decrease the likelihood of creasing in the copper shim and to decrease the likelihood of reduction in elasticity. A quasi-static distributed load via displacement was applied to the suspended mid-section of the sensor, which was enough to bend the sensor so that it touched the lower step, i.e., the sensor’s free end has moved by 1.5 mm. It was then released at a certain velocity in a periodic manner at a certain frequency. The ‘bending’ loading followed approximately a typical sine wave. No actual reaction force due to applied displacement of the sensor was measured in the work.

‘Flicking’ simulates a high impact and high velocity loading. In this case, similar to the ‘bending’ loading, the same stepped surface was used. A strip of 3M’s double-sided copper tape of the same thickness as that of the top step was placed between the top step and the sensor, thus increasing the distance between the bottom step and the sensor by another 1.5 mm (Figure 3b). The purpose of the flexible tape spacer is to allow the sensor to have a greater degree of movement, while reducing the likelihood of creeping. The larger gap between the bottom step and the sensor also allows for a greater displacement angle which assists with a ‘flicking’ motion. Similar to ‘bending’, a sudden load via displacement was applied to the suspended section of the sensor, and then released quickly by sliding off and allowing the sensor to spring back rapidly due to the elastic properties of the structure. The characteristic of ‘flicking’ loading appeared to be a decaying wave as shown in Figure 3c. No reaction force due to cyclic displacement of the sensor because of flicking motion was measured. Note that, in addition to the loading type, the loading frequency was varied from 1 to 5 Hz. Loading or excitation frequency was defined as the number of bending or flicking motion cycles being applied onto the sensor per second.

### 3.4. Experiments and Energy Harvesting Circuits

The experimental setup developed for testing the energy harvesting circuits is shown in Figure 4a. The information captured by the data acquisition (DAQ) card was processed using LabVIEW (National Instruments, Sydney, Australia). The maximum, minimum, peak-to-peak and rms currents (and hence voltage and supply power) were calculated and displayed. The captured data was recorded for post processing and for analyzing waveform characteristics.

Effective energy harvesting circuit is crucial to harness the maximum power output from a piezo sensor at the expense of a small mechanical strain energy. Figure 4b shows the energy harvesting circuit used in this study. The sensor electrodes were attached to a 10 MΩ resistor, which was connected in shunt to a DAQ differential input. A large resistance was used to force as much current as possible into the DAQ to create a more accurate measurement. Hence, one electrode was connected to ground through a combined laminate and body resistance $R_{b}$ of approximately 400 kΩ. For the AC-DC conversion, a full-wave diode bridge rectifier and with a 470 nF capacitor in parallel was connected between the sensor and the load resistor. 1N5711 Schottky diodes (Element14, Sydney, Australia) were used due to its low forward voltage (0.41 V maximum) and satisfactory reverse repetitive peak voltage (70 V maximum). The full wave diode bridge rectifier circuit was used, because researchers have observed it to show high electrical power efficiency, especially in low voltage situations [19].

In this paper, poling was not carried out for the sensors studied. Other researchers have studied the effects of poling on energy harvesting PZT sensors [3]. As a preliminary work, we aimed to investigate the fundamental behaviour of PVDF in energy generation, without applying any external electrical field or heat. While poling might have improved the overall power generation, the trends for the effects of sensor geometry, loading, frequency, and strain on power generation will largely remain similar to those reported in this study and so will the conclusions of the paper. Study of the effects of poling on sensor performance remains as a scope of our future work. Note that all sample preparations and experiments to test the sensor performance were conducted at a room temperature of 23 °C at atmospheric pressure.

## 4. Results and Discussion

### 4.1. Capacitance

As a measure of charge storing capability, the capacitances of the sensors with different geometric configurations were measured using a digital multi-meter. For the purpose of comparison, capacitance values were also estimated using the capacitive circuit model of the sensors [20].

Table 2 summarizes the measured and calculated capacitances of the sensors. Clearly, for small PVDF thickness (t = 0.0254 mm), the calculated capacitance values were considerably larger than the measured capacitances. This is expected because the theoretical calculation does not take into account the impedance of the epoxy joiner. Sensors with a larger surface area showed a higher capacitance, which was to be expected, and sensors with the same surface area but different aspect ratios had very similar capacitances. For much larger PVDF thickness (t = 0.254 and 0.508 mm), the calculated capacitance value is very close to the measured capacitance. This is due to the fact that in the practical sensor the impedance of the epoxy joiner has lesser effect on the overall capacitance. The capacitance can give us an indication of charge decay time of the actuator and hence its energy harvesting capability. For instance, the largest measured capacitance in Table 2 was 0.34 nF for the PVDF sensor with A = 25 mm × 25 mm and t = 0.0254 mm. This capacitance was approximately 2% of the capacitance of the PVDF sample with A = 240 mm × 240 mm and t = 0.027 mm studied in [15]. However, it is noted that the surface area of our sensor is only 1.1% of that of [15], which resulted in much lower capacitance. Therefore, it can be argued that the capacitance generated by the proposed sensor is consistent and reasonable. Results on measured capacitance per unit area as shown in Table 2 follow the similar trend and are consistent to that of [15] as well.

### 4.2. Effect of Sensor Type

To measure the effect of sensor type on the electrical power generation capability, three different sensor types with the same dielectric thickness and surface area were tested under ‘Flicking’ and ‘Bending’ loading at a frequency of 3 Hz. Figure 5 shows the mean current (Irms) and power outputs (Pout) for PB, SB, and U sensors.

For both loading types, clearly the electrical output for parallel bimorph sensor was the highest among all three sensor types. The unimorph sensor exhibited the second highest electrical output. The results are consistent the work of [12]. Since the parallel bimorph sensor showed the highest current and power output, we used this sensor type for the remainder of the experiments, i.e., for parametric studies, which are described in the following sections.

### 4.3. Effect of PVDF Thickness

To measure the effect of PVDF thickness, parallel bimorph sensors with three different thicknesses (0.0254 mm, 0.254 mm and 0.508 mm) were tested for the same surface area under ‘bending’ and ‘flicking’ loading at a frequency of 3 Hz. Figure 6 shows the mean current and power output with respect to the thickness. It is seen that the current (Figure 6a) and power output (Figure 6b) decreases with the increase in dielectric thickness under both types of loading.

The results are comparable with the work of [14] who reported that at an optimal dielectric thickness of the sensor, after which, the stored electrical energy decreases with the increase of dielectric thickness. They found that the optimal piezoelectric thickness to substrate thickness ratio was 1.05 for a fixed (200 µm thick) steel substrate, which was independent of geometric dimensions of the sensor. It was postulated that the maximum power output was achieved before the rigidity of the piezoelectric material had a significant effect on the overall rigidity of the sensor. As can be seen from Figure 6, the power output decreases with the increase of dielectric thickness and the maximum output occurs at substrate thickness of 0.025 mm (first data point in Figure 6). PVDF thicknesses we studied were beyond the optimal ratio, which caused a decreasing trend of power output due to the potential dominant stiffness effect of the dielectric material on the overall stiffness of the sensor. For flexible substrate similar to copper shims used in our study, a sharp decreasing trend of power output with the increase of stiffness of polyvinylidene fluoride-trifluoroethylene (PVDF-TrFE) films was reported in [21]. Therefore, the overall trend of power generation with respect to the dielectric thickness seems reasonably accurate. Note that only three PVDF thickness values were studied in this paper and thus the results obtained are not exhaustive, and a detailed study encapsulating wider range of thickness including thinner PVDF films (<0.025 mm) would be of greater interest to conclusively determine an optimum characteristic of the sensors for future work.

### 4.4. Effect of Aspect Ratio

To measure the effect of aspect ratio (L/W) on electrical power generation capability, three sensors with different ratios (4:3, 25:12, and 3:1) at constant thickness, surface area and configuration were tested at a frequency of 3 Hz for both bending and flicking loads. Mean current and power output with respect to aspect ratio are shown in Figure 7. As can be seen from the figure, power output increases up to a maximum value and then decreases as the aspect ratio increases.

There are two factors to consider when analyzing the power output characteristics: one is the moment, which is the product of the force vector multiplied by the beam length, and the other is the width of the sensor. A longer beam length results in a greater moment, which generates larger electrical energy. On the other hand, with a smaller width, the applied load is distributed across a smaller area at the edge of the bend line, which results in a smaller power output. This would explain the parabolic output characteristics of the sensors (Figure 7). The power increased and reached a peak point, followed by a decreasing trend, as L/W ratio increases. The ideal L/W ratio was found to be about 2:1. The findings are slightly in contrast with that of [16], where they reported a noticeable increasing trend of power output only when the L/W ratio was greater than 3.

### 4.5. Effect of Surface Area

To measure the effect of surface area on electrical power generation capability, sensors with four different surface areas of 100, 225, 400, and 625 mm^2^ with the same thickness and configuration were tested at a frequency of 3 Hz. As can be seen in Figure 8, mean current and power increase with the increase of surface area.

Interestingly, the relationship between power output and surface area is slightly different to that found from the simulation in [16]. The authors reported that the power output was greatest for an area of 400 mm^2^ and then decreased with increasing surface area. They further suggested that this was due to the increased flexural stiffness for larger areas, which would result in less strain for the same applied load. It must be noted that they applied a constant magnitude of loading, whereas our method of loading was aimed at maintaining constant strain. This fundamental difference in loading method could be responsible for the discrepancy in the trends of the results.

### 4.6. Effect of Loading Frequency

To measure the effect of loading frequency on electrical power generation capability, a sensor was tested at frequencies ranging from 1 to 5 Hz, which encompasses equivalent loading frequency on wearable fabrics due to human body movements. All other geometric parameters of the sensor were kept constant. Figure 9 shows the test results for different loading frequencies.

The effect of loading frequency had a similar effect to that of surface area for “bending”, the output current increased modestly with frequency up to a point (4 Hz) and then dropped slightly at higher frequencies. The power output increased slightly with frequency up to point (4 Hz) before dropping marginally at higher frequencies. With “flicking”, the current and power output increased almost linearly with frequency all the way up to 5 Hz. The increases were much more pronounced than seen in the case of “bending”.

The trends can be explained by carefully observing the instantaneous current waveforms for 3, 4, and 5 Hz, as can be seen in Figure 10. When the sensor was stressed, the current spiked initially and then discharged. This is consistent with the observations of [13], although the force application is slightly different. The discharging of a sensor can be characterised by the time constant τ, which indicates how quickly the sensor is able to discharge. The time constants for both ‘flicking’ and ‘bending’ loading at 3–5 Hz were calculated by analyzing the data associated with the instantaneous waveforms shown in Figure 10. There was a distinct time gap between the positive and negative currents for the “up” and “down” motions with ‘flicking’, whereas the two regions were much closer together for ‘bending’. In the case of “bending”, when the loading was done at 3 Hz, the induced current had enough time between each ‘down’ and ‘up’ motion to discharge. Once the loading frequency reached 5 Hz, there was not enough time for the current to discharge naturally. Therefore, potential energy from that motion was not completely transferred into electrical energy. This led to the logarithmic increase in power output with the increase of frequency. Indeed, the power even dropped at 5 Hz due to this effect. As Figure 10 demonstrates, with the “Flicking” motion, the loading was much more sudden and the induced current had ample time to return to 0 before the next load is applied, so the ‘pulse-cutting’ was not seen.

As summarized in Table 3, the time constants for 3 Hz and 4 Hz of ‘bending’ were larger than that for 5 Hz. This reflects the fact that the 3 Hz and 4 Hz pulses are longer than the 5 Hz pulse, and are able to impart a greater amount of energy per pulse. For all frequencies, the time constants for ‘bending’ were consistently greater than that for ‘flicking’. The results are in line with the findings of [13] which reported that an increase in static pressure or stress in PZT sensor results in a decrease in decay time constant, and thus suggest that ‘flicking’ imparts greater pressure than ‘bending’. Moreover, the mechanical stability of the flexible electrodes can be compromised under both loading types when subjected to large number of cycles of operation. Eventually, this can cause permanent deformation of the sensor device, which can affect the overall power generation efficiency of the sensor [22]. This warrants further investigation and remains within the scope of our future work.

### 4.7. Effect of Mechanical Strain

To measure the effect of strain on electrical power generation capability, two sensors with the same PVDF thickness (of 0.0254 mm) and two different surface areas (of 400 and 625 mm^2^) were tested with different loading in terms of maximum displacements of PVDF sensor end at 1.5, 3.0, 5, and 6.5 mm, which corresponds to strains of 0.00602, 0.0122, 0.0209 and 0.0283 for the 400 mm^2^ sample, and 0.00603, 0.0123, 0.0216, and 0.0303 for the 625 mm^2^ sample. The values of strains are chosen based on human body movements at key flex points such as the inside of the elbow and behind the knee. The strain was calculated using a simple formula to find the angle of separation between the bent sensor and the bottom step when fully loaded, which is as follows.
(1)$$∆l=2\pi t_{s}\left(\frac{\theta }{360}\right)$$
where θ is the angle of separation and, ∆l is the change in length, and $t_{s}$ is the total sensor thickness. Figure 11 shows the mean current and power output with respect to mechanical strain.

It can be seen from the figure that for the sensor with surface area of 400 mm^2^, clearly, mean voltage and current increased when the strain increases from 0.00602 to 0.0122 (first two data points on the graphs). They, however, did not increase as rapidly beyond past the strain of 0.0122. For the sensor with surface area of 625 mm^2^, the voltage and current curves started to flatten after the strain of 0.0216. The limiting strain seems to be higher for sensor with larger surface area. Our results suggest that an upper limit exists on the amount of deformation a material may experience before reaching its optimum charge generating capability. Defects in the sensor construction may also limit the maximum strain for power generation. With a large deformation, the dielectric material may become detached from the conducting substrate, hence increasing the impedance of the gap between the two materials and decreasing the energy generating potential. Therefore, a more thorough analysis would be required to investigate the full effect of this.

### 4.8. Analysis of Energy Harvesting Circuit Efficiency

To test the efficiency of the energy harvesting circuit, the electrical output parameters were measured for different system configurations with a 625 mm^2^ sensor with 3 Hz ‘Bending’ loading at strains of 0.0123 and 0.0216. Table 4 summarizes the test results on the performance of the energy harvesting circuit that was presented in Figure 4b. As the capacitors take a certain amount of time to charge as determined by the RC time constant, $\tau_{RC}=R_{l}C_{l}$, where $R_{l}=10\;MΩ$ and $C_{l}=470\;\mathrm{nF}\;\mathrm{and}\;1\;\mu F$, the measurements could not be taken from the start of the loading, but rather taken from the time at which the capacitor became fully charged. The charging time was determined by collecting data from a “cold start” and determining the point at which the charge stopped increasing in post-analysis as shown in Figure 12 (for ε = 0.0216). As less strain generates less current, the charging time for the capacitors was higher for the lower strain measurement (ε = 0.0123). After the capacitor is fully charged (the time of which is dependent on the energy of the signal being fed into the capacitor) we would expect the *I_rms_* output to hold steady under continuous loading of the same frequency and magnitude. This concept was reiterated in [12] for energy.

However, what actually occurs is that after the current stops increasing due to the capacitor’s energy storage capacity, it starts to decrease, as shown in Figure 12. It may be due to some parasitic element of the capacitor itself, such as leakage resistance or current leakage through the rest of the system. It is clear that once the capacitor reaches its maximum energy storage capability, the current is being drained faster than it is being produced under constant loading.

As shown in Table 4, at ε = 0.0123, the power generated at the output of the rectifier (refer to Figure 4b) was slightly higher than the power generated with no rectifier. As the values were very close, we can put this down to small changes in the applied force. For ε = 0.0216, the power generated at the output of the rectifier is less than the power generated with no rectifier, which is what we would expect due to the voltage drop and associated power dissipation through the diodes.

As Table 4 shows, the greatest constant DC current generated in this study was 34.7 nA corresponding to a generated power of 12 nW when the load capacitor was 470 nF. If such sensor device was used to charge a typical 3.7 V, 1000 mAh phone battery from flat with a 100% efficient voltage step up circuit, it would take approximately 307 million hours to charge fully. Charging a 2.8 V, 1000 mAh pacemaker battery would take 233 million hours.

The largest AC current generated was 49.4 nA with a corresponding power output of 24.5 nW. However, the power output reduces to almost half for the same strain (0.0216) when a full-wave rectifier is used with a 470 nF capacitor. This power output capability is far lower than that predicted (1800 µW) in [16], and the theoretical recoverable power of 0.33 W from upper arm motion calculated by [6]. It is however to be noted that the simulated PVDF patch in [16] had a much larger surface area (1600 mm^2^ as opposed to 625 mm^2^ in our study), which should explain the reduced power generated in our case.

The main reason behind our generator’s poor power generation capability is likely to be the absence of poling in the test procedure. Noise in the system was a big issue during testing. Table 5 displays the different earthing configurations with the associated noise levels. Figure 13a shows direct grounding of the sensor, while Figure 13b shows grounding through the body. The sensor used was a 20 × 20 mm parallel bimorph with 0.0254 mm thick PVDF. Clearly, the best configuration to minimize system noise is with the sensor connected to ground through the body. The most noise occurs when the sensor is connected to the body with no grounding at all (shown as “Direct” in Table 5). As the sensor is unlikely to remain completely out of contact from the human body, the best solution is to ground the body. For future applications, this may require a ground point integrated into the clothing or grounding directly from the electronic device, which is being charged.

## 5. Conclusions and Outlook

In this paper, the effects of PVDF sensor type, geometry, loading frequency and strain on the electrical power generation capability were studied. Two types of movements were emulated, namely ‘bending’ and ‘flicking’. The major findings from the study are summarized as follows.

Among the three sensor configurations, parallel bimorph sensor was shown to be the best energy harvester in that it generated the highest amount of electrical power. Practically measured capacitance values for PVDF sensors with large thicknesses were found to be close to the theoretical values [23].Given the range of PVDF thickness studied, power output seems to decrease with increasing PVDF thickness. Such trend could be because the stiffness of the PVDF dielectric might have dominated power generation, and the sensor thickness passed the optimum value after which the power output decreased.Power output increased consistently with the increase of sensor surface area. Flicking movement showed greater power output than bending for all surface areas as well as far greater increase in power output with surface area. As for the effect of geometry, power output increased initially with increasing aspect ratio, reaching the maximum value at an aspect ratio of around 2:1, and then decreasing for higher aspect ratios.Mean current and power output showed a significant increase with the increase in loading frequency for flicking. However, for bending, modest increases were noticed initially (3 Hz and 4 Hz) with the output dropping for higher frequency (5 Hz).Power output also followed an increasing trend with the increase of mechanical strain up to a certain value of strain. In addition, an increase in loading frequency resulted in an increase in power generation, however, it did not increase linearly as “wave-cutting” decreased the potential for charge generation at higher frequencies.A simple energy harvesting circuit was found to successfully convert the AC signal generated by the sensor into DC signal, but at the cost of a significant loss in power output.

The above findings for PVDF sensors were largely in line with the trends observed using sensors made from other piezoelectric materials, for example, PZT. Although the PVDF sensors studied were not able to generate significant power output to be readily useful in practical applications, some credible relationships and insights on the performance of PVDF sensors were obtained. To realise the full potential of PVDF sensors, future work will focus on more precise preparation of the PVDF sensors with poling, increasing the efficiency of the energy harvesting circuit, optimizing the geometry of the parallel bimorph configuration, and deploying efficient noise reduction techniques.

In this study, the resistance load was kept constant for all parametric analysis of the sensors. However, the change in resistance load in the energy harvesting circuit has a significant influence on the current and power output. Using a coupled piezoelectric circuit-finite element method (CPC-FEM) on vibration-based PZT energy harvesting devices, Zhu et al. [24] reported that when the resistive load within the circuit increases, power output increases up to a peak (maximum) point, and after which, decreases. On the other hand, vibrational amplitude decreases and then picks ups. They found that maximum power occurs at a resistive load of 488 kΩ and vibrational amplitude of 100 µm. The findings imply that the electrical power generation is dependent on the resistive load. Frequency and phase response of the output current and power with respect to the changing resistive load should be analysed to better understand, explain and optimize the power generation capacity of the PVDF sensor [25]. In addition to noise reduction via grounding through the body, the variation of resistive load must be taken into account to elude the efficiency of the energy harvesting PZT sensors. Furthermore, our future work will aim to address theoretical background and simulation to reveal the efficacy of the sensors. Though, in this study, all experiments were conducted in ambient temperature, the efficiency of power generation can be impacted by the change in temperature. For instance, a recent study by Bernard et al. [26] reported the increase of dielectric constant of the PVDF with the increase of temperature from 30 °C to 60 °C and strain up to 2.5%, which caused a decrease in efficiency down to 60–75%, as opposed to ideal conversion efficiency of 90%. They suggested that the circuit output capacitance must be decreased in proportion to be adapted to operation conditions in order to restore higher power generation capacity.

## Figures and Tables

**Figure 1 materials-14-01895-f001:**
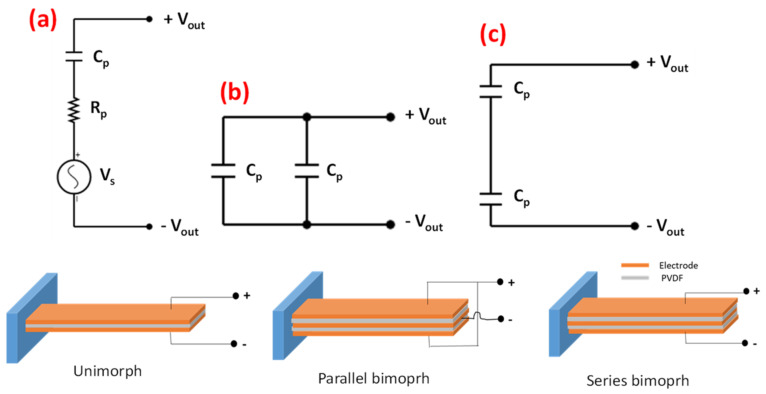
Voltage generator piezoelectric circuit model (**a**) unimorph (**b**) parallel bimorph (**c**) series bimorph [12].

**Figure 2 materials-14-01895-f002:**
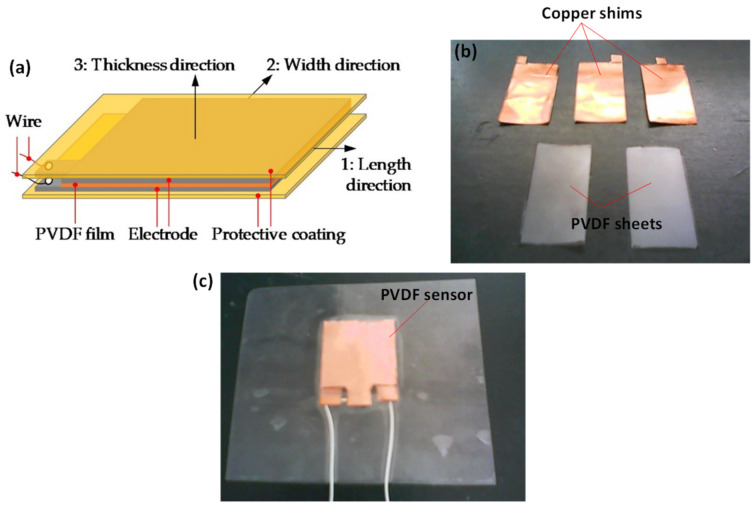
(**a**) Schematic of a typical sensor design (**b**) PVDF and copper sections before being joined and (**c**) laminated PVDF bimorph sensor.

**Figure 3 materials-14-01895-f003:**
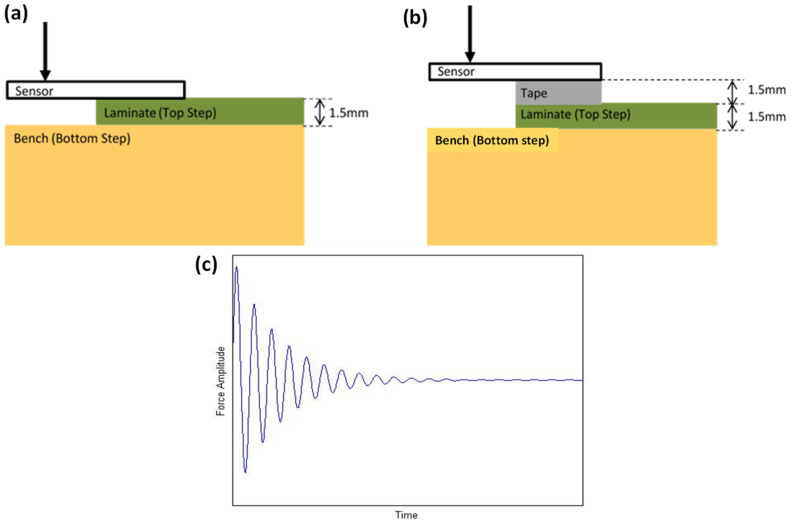
(**a**) Cross sectional view of “bending” loading setup and (**b**) cross sectional view of “flicking” loading setup (**c**) “Flicking” loading representation of force over time.

**Figure 4 materials-14-01895-f004:**
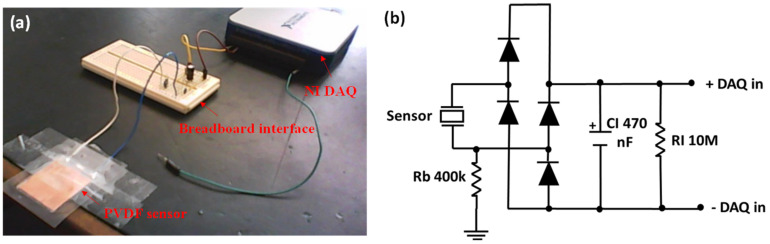
(**a**) Experimental setup for sensor testing and (**b**) energy harvesting circuit with a 470 nF load capacitor.

**Figure 5 materials-14-01895-f005:**
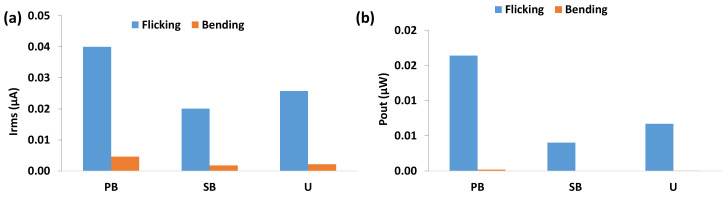
(**a**) Mean current and (**b**) power output for parallel bimorph (PB), series bimorph (SB) and unimorph (U) sensors.

**Figure 6 materials-14-01895-f006:**
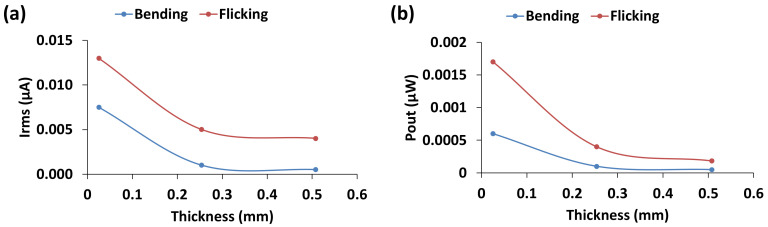
(**a**) Mean current and (**b**) power output for increasing PVDF thickness.

**Figure 7 materials-14-01895-f007:**
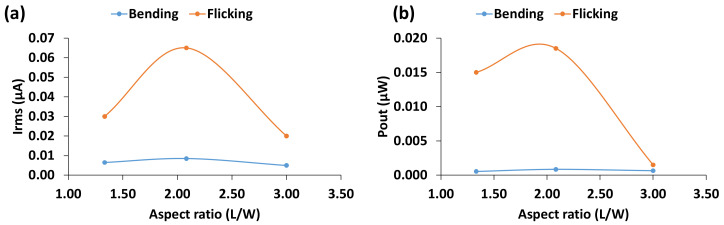
(**a**) Mean current and (**b**) power output with respect to aspect ratio (L/W).

**Figure 8 materials-14-01895-f008:**
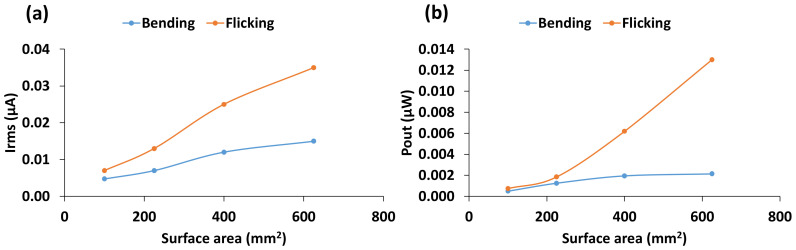
(**a**) Mean current and (**b**) power output with respect to surface area.

**Figure 9 materials-14-01895-f009:**
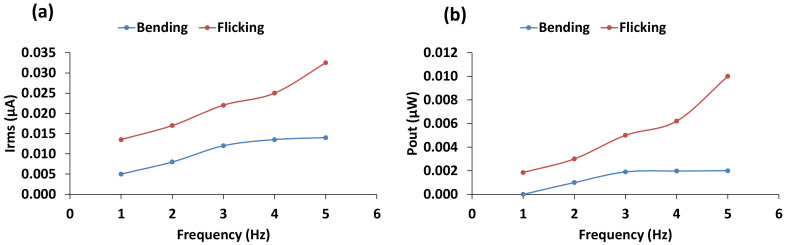
(**a**) Mean current and (**b**) power output with respect to loading frequency.

**Figure 10 materials-14-01895-f010:**
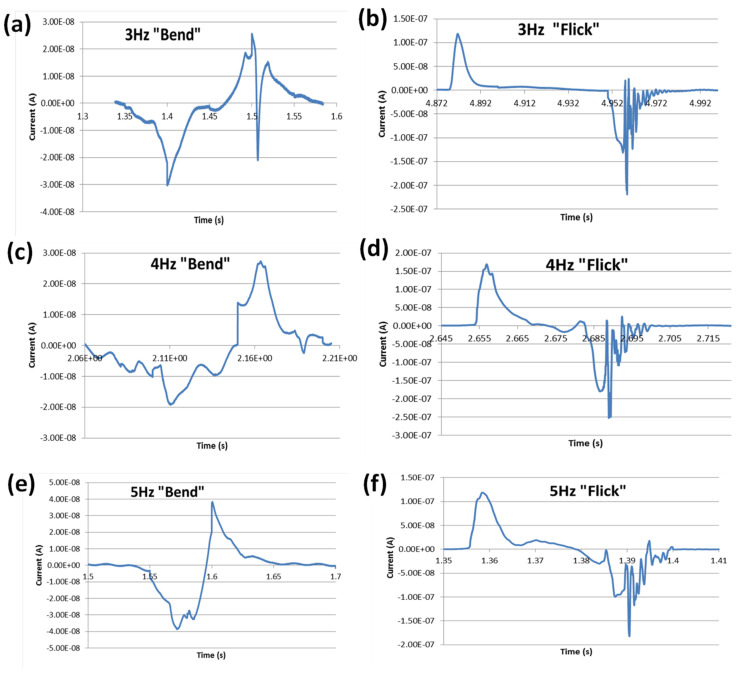
Current waveforms with respect to loading frequency for “Bending” (**a**,**c**,**e**) and “Flicking” (**b**,**d**,**f**).

**Figure 11 materials-14-01895-f011:**
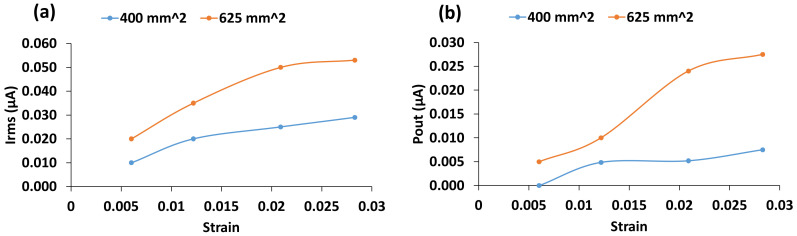
(**a**) Mean current output (**b**) mean power output with respect to strain.

**Figure 12 materials-14-01895-f012:**
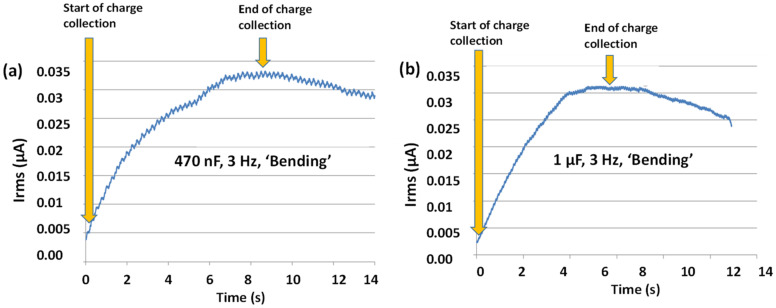
Current over time for constant force loading for (**a**) 470 nF and (**b**) 1 µF capacitors.

**Figure 13 materials-14-01895-f013:**
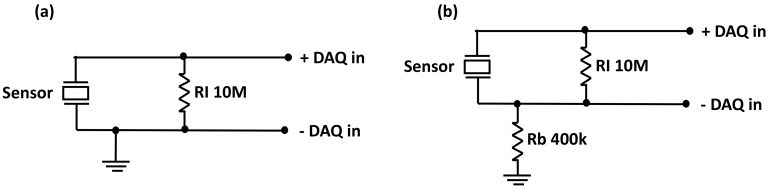
(**a**) Circuit grounding directly to the sensor and (**b**) circuit grounding through the body.

**Table 1 materials-14-01895-t001:** Sensor types and geometries studied.

Sensor Type	Thickness, t (mm)	Aspect Ratio, L/W	Surface Area, A (mm^2^)
PB	0.0254	1	100
PB	0.0254	1	400
PB	0.0254	1	625
PB	0.0254	25:12	300
PB	0.0254	4:3	300
PB	0.0254	3	300
PB	0.254	1	225
PB	0.508	1	225
PB	0.0254	1	225
U	0.0254	1	225
SB	0.0254	1	225

**Table 2 materials-14-01895-t002:** Capacitance results for different sensor geometries and circuits.

Sensor Type	Thickness t (mm)	Aspect Ratio L/W	Surface Area, A (mm^2^)	Calculated Capacitance (nF)	Measured Capacitance (nF)	Measured Capacitance per Unit Area (pF/mm^2^)
PB	0.0254	1:1	100	0.6	0.06	0.6
PB	0.0254	1:1	225	1.4	0.1	0.44
PB	0.0254	1:1	400	2.5	0.3	0.75
PB	0.0254	1:1	625	3.9	0.34	0.54
PB	0.0254	25:12	300	1.9	0.25	0.83
PB	0.0254	4:3	300	1.9	0.34	1.13
PB	0.0254	3:1	300	1.9	0.23	0.77
PB	0.254	1:1	225	0.14	0.12	0.53
PB	0.508	1:1	225	0.071	0.09	0.4
U	0.0254	1:1	225	0.71	0.16	0.71
SB	0.0254	1:1	225	0.36	0.1	0.44

**Table 3 materials-14-01895-t003:** Time constants for increasing load frequency.

Loading Frequency (Hz)	Time Constant (τ)	
	Bending	Flicking
3	0.0157866	0.00376225
4	0.01664428	0.0044029
5	0.00989307	0.00300784

**Table 4 materials-14-01895-t004:** Electrical output characteristics with and without energy harvesting circuitry.

Strain	Rectifier	Capacitor (nF)	* *I_pp_* (nA)	*I_rms_* or ^$^ *I_dc_* (nA)	*P_out_* (nW)	Charging Time (s)
0.0123	No	-	135	34.0	11.0	-
0.0216	No	-	294	49.4	24.5	-
0.0123	FW	-	116	35.6	12.8	-
0.0216	FW	-	140	47.6	22.7	-
0.0123	FW	470	2.88	26.5	7.01	9.92
0.0216	FW	470	4.64	34.7	12.0	7.34
0.0123	FW	1000	1.42	20.3	4.12	13.91
0.0216	FW	1000	1.93	33.3	11.1	11.42

* AC current with no rectifier circuit, ^$^ DC current with rectifier circuit.

**Table 5 materials-14-01895-t005:** Noise measurements for different connections.

V_peak_	V_rms_	Grounding	Sensor
2.37 mV	2.16 mV	Through Body	Yes
75.8 mV	53.1 mV	Direct	Yes
3.66 mV	3.01 mV	Ungrounded	Yes
4.94 mV	4.12 mV	Through Body	No
88.7 mV	61.3 mV	Direct	No
10.1 mV	6.96 mV	Ungrounded	No

## Data Availability

Data sharing is not applicable to this article as this is an on-going project.

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
