# Peer review of "Evaluating Energy Generation Capacity of PVDF Sensors: Effects of Sensor Geometry and Loading"

_materials, 2021, doi:10.3390/ma14081895_

Round 1

Reviewer 1 Report

Dear Authors,

your article is devoted to a relevant topic, the study of piezoelectric harvesters. The article contains a large amount of experimental data. The presented results will undoubtedly be of interest to the international scientific community. At the same time, from the point of view of the reviewer, the article needs to be improved. It is worth making significant additions to the article.

  1. It is necessary to describe in more detail the conduct of experiments in terms of the load and its type on the piezoelectric element: displacement amplitudes, duration of exposure. How is the mechanical load on the piezoelectric element of the harvester ensured: on the entire surface or concentratedly - along the edge? Is it all over the edge? How much force was applied? How was the excitation frequency set at 2, 3, 4, 5 Hz (line 202)?

When describing the experiment, ambiguous concepts were found, for example:

line 198 "A LITTLE force was used ...". How small? Small compared to what?

2.The authors describe the effect of thickness, geometry, frequency action on the piezoelectric element. However, when analyzing the output signals of the piezoelectric harvester, the relationship between the frequency response and phase response of these signals with the volume of the piezoelectric and the properties of the material of the piezoelectric element (primarily the piezomodule) is not evaluated. Although in extensive reviews and monographs (for example, A. Erturk, D.J. Inman. Piezoelectric Energy Harvesting, John Wiley & Sons, Ltd., 2011 DOI: 10.1002 / 9781119991151), methods for describing this relationship are presented both theoretically and practically for various piezoelectric materials.

  1. More attention should be paid to the description of the experimental methodology:

– what material the tape was made of in Fig. 3c? It is not possible to find an unambiguous answer to this question in the text of the article. Does the choice of material affect the bending behavior of the piezoelectric element under ‘flicking’ load? Is the variation in step height permissible in order to be considered a ‘flicking’ load?

– why did the authors choose unimorph (U), parallel bimorph (PB) and series bimorph (SB) in cantilever design? The article does not mention this. Designs, for example, such as an umbrella or a mushroom of harvester, are not the worst opportunities for the weekend. Designs, for example, such as an umbrella or a mushroom, provide not the worst possible output power

– it can be assumed that the measurement results are highly dependent on the ambient atmosphere and temperature. It is worth specifying this parameter. How will the results change with increasing or decreasing temperature and environment? How to minimize this dependency?

It can be assumed that the measurement results depend significantly on the humidity of the surrounding atmosphere and temperature. It is necessary to specify the values of these parameters when making measurements. How will the results change with increasing or decreasing ambient temperature and humidity? How to minimize this dependency?

Author Response

Please see the attachment for detailed reply and rebuttal. 

Reviewer 2 Report

This study provided useful insight and experimental validation into the optimization process for an energy harvester. This study is meaningful to the community and field. This paper can be published after considering the following comments:

  • The authors should compare the differences between PVDF made transducer and piezoelectric ceramic made transducer.
  • What are the best parameters for an optimal transducer?
  • How did you deal with noise and electrical mismatch in the experiments?
  • The authors are suggested to cite previous similar study: Trends in Biotechnology 34(5): 420, 2016; European Radiology 28(5): 2176, 2018.
  • Will this technique be applied in humans in wide-scale?

Author Response

(The authors gave the same response as above.)

Round 2

Reviewer 1 Report

Dear Authors,

I have read your answers to my questions and wishes. Your answers satisfied me.

Respectfully,

Reviewer